# Traditional Diet and Environmental Contaminants in Coastal Chukotka II: Legacy POPs

**DOI:** 10.3390/ijerph16050695

**Published:** 2019-02-27

**Authors:** Alexey A. Dudarev, Valery S. Chupakhin, Sergey V. Vlasov, Sveta Yamin-Pasternak

**Affiliations:** 1Department of Arctic Environmental Health, Northwest Public Health Research Center, 191036 St-Petersburg, Russia; valeriy.chupakhin@gmail.com; 2Northwest Branch of Research and Production Association “Typhoon” (RPA “Typhoon”), 199397 St-Petersburg, Russia; 18vsv49@gmail.com; 3Department of Anthropology, Institute of Northern Engineering, University of Alaska, Fairbanks, AK 99775, USA; syamin@alaska.edu

**Keywords:** subsistence food, traditional diet, Indigenous people, environmental contaminants, PTS, POPs, DDT, PCB, food safety limits, coastal Chukotka, Russian Arctic

## Abstract

The article is the second in the series of four that present the results of a study on environmental contaminants in coastal Chukotka, conducted in the context of a multi-disciplinary investigation of indigenous foodways in the region. The article presents the results of the analysis of legacy Persistent Organic Pollutants (POPs) found in the samples of locally harvested food and indoor matters, collected in 2016 in coastal Chukotka. Temporal trends and circumpolar comparisons of POPs in food have been carried out. Estimated daily intakes (EDIs) of POPs by local food consumption were calculated based on the food intake frequencies (questionnaire data). Concentrations of the studied legacy POPs in marine mammal blubber were relatively high (up to 100–200 µg/kg ww) but not exceeding the allowable limits. Gray whale blubber and whale mantak were the most contaminated foods, followed by the ringed, spotted and bearded seal blubber, then by walrus blubber and fermented walrus (deboned walrus parts aged in subterranean pits, typically over a period of 6 months). At the backdrop of general decrease or invariability (compared to the previous coastal Chukotka study 15 years ago) of the majority of POPs, an increasing tendency of HCB, mainly in marine mammals, were noted. Legacy POPs in marine mammals sampled in Chukotka were generally much lower than in those sampled in Alaska and northern Canada. We suggest that the Alaska Coastal Current from the Bering Sea plays a major role in this phenomenon. Analyses of the additional sources of in-home food contamination (home-brewed alcohol, domestic insecticides) have revealed relatively high levels of HCHs, DDTs and PCBs, which still represent a share of dietary exposure of local people to POPs.

## 1. Introduction

### 1.1. Overview

Many Persistent Organic Pollutants (POPs) are polyhalogenated hydrocarbons (in this article only polychlorinated compounds, the so called legacy POPs, are considered); some of them are listed in the Stockholm Convention as the most dangerous organic compounds banned from production, use, transportation, and are subjects to extermination. All POPs are xenobiotics, but not all are products of deliberate chemical synthesis due to which effective (but hazardous for the environment and human health) pesticides have been produced. Such pesticides (e.g., dichlorodiphenyltrichloroethane–DDT) and technical additives (e.g., polychlorinated biphenyls–PCBs) have been used for decades in agriculture and industry, circulated and accumulated in the planet environment. POPs are partially entering the Earth biosphere as by-products of various industrial production processes, with emissions from fossil fuel power plants, waste incineration, fires, etc. [1].

Some POPs from List A (prohibited production and use) of the Stockholm Convention are still produced in several countries, particularly in China (despite ratification of the convention) and are exported, and many countries have stocks or imports of POP-containing pesticides, and are still applying them in agriculture. A significant number of countries in Africa, South-East Asia and Latin America continue to use DDT (list B “restricted use”) to control malaria vectors and other vector-borne diseases. Extremely high incidence rate of malaria in some southern countries still leads to millions of victims, while no effective alternative to DDT has been developed to date. The majority of legacy POPs are characterized [1] by a combination of several specific features:They are stable in the environment, resistant to thermolysis, photolysis, hydrolysis, chemical and bacteriological degradation and, as a result, persistent (for years and decades);They can be transported globally (over thousands of kilometers) due to their the specific physical and chemical properties (primarily, low solubility in water and volatility). Long-range transport of POPs to the Arctic occurs with atmospheric fluxes (warm air masses from regions of low and middle latitudes are precipitated by collision with cold Arctic air fronts), river, sea and oceanic currents;They undergo to bioaccumulation, reaching high levels in the highest echelons of food chains due to the processes of biomagnification, when the concentrations of POPs in organisms on the top of food pyramids are thousands-millions times greater than those in air, soil, water and organisms of the bottom of food pyramids;They show to have extremely slow metabolism and withdrawal from the human body, which contributes to the progressive (over many years) accumulation of significant doses of POPs even when their concentrations in food are insignificant, but when the oral exposure is constant or has a regular seasonal periodicity;They are toxic and dangerous to biota and human health in low (and even ultra-low) concentrations. Lipophilicity of POPs contributes to their accumulation in adipose tissues and in lipid-rich organs (including brain and spinal marrow). High permeability of the blood-brain barrier for many POPs predetermines the risk of direct toxic effects of POPs on the central nervous system and endocrine sphere. The increased content of POPs in the body can provoke serious health disorders, especially in the neuroendocrine, immune and reproductive functions, fetal development and antitumor resistance. Some POPs are “hormone imitators” (“hormone-like xenobiotics”), inhibiting the production of natural hormones in the body and disrupting the normal course of processes regulated by the endocrine glands, such as spermatogenesis, ovulation, and the forming of hormonal imbalances and immunosupression. Most of POPs easily penetrate the placental barrier, entering the baby’s body during fetal development and in infancy with mother’s milk.

The diet of Indigenous peoples of the Arctic is based mainly on local foods, which are also consumed to varying extent by non-indigenous Arctic residents. The dietary route remains the main way of exposure to POPs of the Arctic population.

Presence of POPs in the food chains in the circumpolar Arctic is due mainly by global transfer. Low concentrations of POPs are inherent to wild plants, berries, mushrooms, marine crustaceans, mollusks and other invertebrates, and the species of fish that are characterized by a low fat content. POPs can accumulate in “fatty” species of fish (e.g., halibut, herring, catfish, some salmon species), birds (e.g., various ducks) and, above all, marine mammals (polar bears, cetaceans, pinnipeds). The greatest natural accumulator of POPs in Arctic environment is the blubber of marine mammals as they are on the top of marine food chains and due to bioaccumulation of lipophilic contaminants. Thus the Indigenous people who reside in the coastal communities of circumpolar Arctic and consume marine mammals have higher exposure to POPs (and correspondingly enhanced health risks) compared to people in inland communities.

### 1.2. Circumpolar POPs Trends in Local Foods, a Brief Summary

During last decades the studies on local food contamination by POPs and dietary human exposure to POPs have been carried out in Alaska, Canada, Greenland, Faroe Islands, Scandinavian countries, and Russia. The summarized results of these long-term (1990s—2015) multi-national studies are presented in the consistent Reports of the Arctic Monitoring and Assessment Program (AMAP) [2,3,4,5,6] and in Canadian Arctic Contaminants Assessment Reports of the Northern Contaminants Program [7,8,9,10].

Vast amounts of literature data show that most of the legacy POPs appear to be declining in the Arctic environments during recent decades, including biota from marine, freshwater, and terrestrial ecosystems. Meta-analysis was applied to 316 time-series of legacy POPs in Arctic biota with the purpose of generating a temporal trend data collected over the past two to three decades for locations from Alaska in the west to northern Scandinavian in the east [11]. Most of the analyzed time-series of legacy POPs showed decreasing trends; only few time-series showed the increasing trends. α-HCH, γ-HCH (isomers of hexachlorocyclohexane) and ΣDDT had a relatively high proportion of time-series showing significantly decreasing trends; ΣCHL (sum of chlordanes) had the lowest proportion. β-HCH was an exception, where long-range transport through the ocean, and not the atmosphere, may explain several increasing trends that were detected in the Canadian Arctic [11].

Most of the legacy POPs appear to be declining in ringed seals and polar bears. There is, however, strong evidence over the last decade of stalling declines in PCB and CHL levels [12], which may be related either to continued emissions from in-use materials and waste sites, despite the banning of these products in the 1970’s or to the release of these compounds archived earlier in local or global environmental reservoirs (soils, vegetation, snow, ice) due to climate change (e.g., warming, permafrost melting, forest fires) and subsequent redistribution [13].

Summarized multi-year Canadian Arctic data on food POPs contamination and identification of the trends was presented in the NCP Report [14]. The declining trend in concentrations in biota is most apparent for organochlorine pesticides and less evident for PCBs and chlorobenzenes (including HCB). Declines of legacy POPs have been generally more rapid in freshwater fish than in marine animals. Declines were seen for ΣHCH, ΣCHL, ΣDDT and toxaphene in lake trout of ≥5%/year and in seabird eggs and marine mammals <5%/year [14].

Overall, there are declining trends in Canadian Arctic marine mammals with the relative magnitude of ΣDDT > α-HCH > Σ10PCBs > ΣCHL. These trends are generally similar to those in Arctic marine mammals in Greenland and Svalbard. ΣHCH declined in seals, beluga and polar bears due to rapid decline of the major isomer α-HCH. However, β-HCH, the more bioaccumulative isomer, increased in the same species. This increase in β-HCH in seals varied regionally, and this case highlights the importance of ocean water moving through the Arctic archipelago from the Pacific Ocean via the Bering Sea and possibly Russian freshwater inputs [14].

PCBs and ΣCBz had also increased in burbot and lake trout in Great Slave Lake in the period 2001–2005. These increases were not seen in lake trout in the Yukon or in landlocked char. Annual sampling made it possible to observe these changes. These temporary increases suggest some process that is influencing the availability of POPs in the Mackenzie basin. Climate warming has been suggested, however a general warming trend would not explain the increase followed by a decrease in concentrations. Nevertheless, shifts in the burbot and lake trout diet and feeding areas, which could also be induced by climate change, might affect the presence of contaminants. Other possibilities include mobilization of legacy sources due to warming, e.g., increased erosion of river sediments [14].

It is important to consider that concentrations of contaminants in higher trophic levels (e.g., bears, seals) are strongly mediated by local or regional food web structure. Expansion of species distribution and invasion of southern species further north, due to a warming climate, has the potential to alter food web structure (alternative prey may have higher or lower levels of contamination) and, subsequently, the contaminant accumulation in marine mammals across the Arctic. Variation in the body condition could alter contaminant burdens in individual animals. Periods of reduced nutrition and increase in bodily stress (e.g., during periods of fasting influenced by climate change) may result in increased lipid catabolism, which may mobilize and release contaminants into circulation [13].

### 1.3. Objectives and Tasks of the Follow-up Study in Coastal Chukotka

Collection and chemical analysis of local foods for POPs were conducted in coastal Chukotka in 2001–2002, and other regions of the Russian Arctic, as part of a larger Russian Arctic PTS study [15]. The community-based dietary and lifestyle survey (interviews of 251 Indigenous people in Uelen settlement) based on self-reported daily (weekly, monthly) food frequencies were carried out at same period. The study has revealed the coastal Chukotka to have the highest (compared to other Russian Arctic regions) levels of POPs contamination found in local foods and the highest POPs exposure levels in local Indigenous people. The comprehensive article devoted to the dietary exposure to POPs in Chukotka Native people (based on the 2001–2002 collected data) was published in 2012 [16].

The objective of the present part of the project is to assess the environmental legacy POPs in subsistence species (including fish, terrestrial and marine mammals), compare the results with those obtained in coastal Chukotka 15 years ago (the follow-up) and estimate the daily intakes (EDIs) of the different legacy POPs with different food items by the local Indigenous people.

Another important task on which we report here is the assessment of the prevalence of household insecticides use in Chukotka and the analysis of POPs content in the common present-day household insecticides. Information obtained through the previous study in 2001–2002 has demonstrated wide use of insecticides in Chukotka: in homes (including kitchens, places of storage and preparation of food), canteens, kindergartens, schools, medical institutions, etc. Samples of some aerosols, crayons and gels (for combating cockroaches) we have tested 15 years ago (both Chinese and Russian production), contained considerable concentrations of POPs (including DDTs, HCHs, PCBs) which when contacting with food (kitchen walls washouts were also contaminated), were capable of additional food contamination. Also among the tasks is a follow-up analysis of POPs in home-brewed alcohol, which showed significant contamination of the samples analyzed 15 years ago [16].

## 2. Materials and Methods

### 2.1. Field Sampling

Collection of samples of local foods, home-brewed alcohol and indoor matters was carried out in all three study settlements (Enmelen, Nunligran and Sireniki). Samples of fish (marine, migratory, freshwater), meat of terrestrial mammals (reindeer, hare), meat and blubber of marine mammals (whale, walrus, seals) have been collected. Most of the samples have been kindly provided by the local people; the authors collected a few samples of fish and hare by ice-fishing and hunting near Enmelen during the 2016 fieldwork.

Total 37% of samples have been pooled; each pool consisted of several specimens of each biological species similar in age (size). Approximately equal in weight (size) pieces of the same type of tissue (organs) were selected for pooling. When pooling the fish species, 5–7 specimens were selected, the sizes of which were typical average for these species; a piece of muscle tissue was cut from the central part of the specimen back (up to the backbone, without affecting it), and packed together with several other pieces selected from different specimens of the same species into one package. Hare meat was taken from the muscles of the lower limbs, poultry meat—from pectoral muscle, marine mammal meat (whale, walrus, seal, etc.)—from the *latissimus dorsi* muscle. After pre-treatment, packing, marking and freezing, all samples were delivered to St-Petersburg in thermo-containers, which help prevent their thawing during transportation.

Washouts of indoor areas (walls) were limited to dry non-porous surface squares 400 cm^2^ (20 × 20 cm) using sterile medical cotton wool tampons (medical, sterile, bleached without chlorine) soaked with hexane (grade 1, pure super, UV absorption 0.5 optical units/cm for 200 nm). Each surface was wiped twice and the 2 tampons were wrapped into aluminum foil and put into a plastic container with screw top (for biological samples, EN 829). The total number of samples collected and analyzed for POPs was 57 (Table 1), among them 16 samples of fish, 28 of marine mammals, six from land mammals, one bird, two of home-brewed alcohol, two of insecticide gel, and two wash-outs.

### 2.2. POPs Analyzed in the Collected Samples

The analyzed organochlorines include: a-HCH, b-HCH, g-HCH and ∑HCHs; heptachlor, heptachlorepoxide, *trans*-chlordane, oxychlordan, *cis*-nonachlor, *cis*-chlordane, *trans*-nonachlor and ∑CHL; *o,p’*-DDE, *p,p’*-DDE, *o,p’*-DDD, *p,p’*-DDD, *o,p’*-DDT, *p,p’*-DDT and ∑DDTs; mirex, tetraclorobenzenes, pentachlorobenzene, hexachlorobenzene (HCB); PCB#28, PCB#31, PCB#52, PCB#99, PCB#101, PCB#105, PCB#118, PCB#128, PCB#138, PCB#153, PCB#156, PCB#170, PCB#180, PCB#183, PCB#187 and ∑PCB15.

### 2.3. Chemical Methods and Laboratory Equipment Used for POPs Analysis

Chemical analyses of POPs in all samples were performed by the Northwest branch of Research and Production Association “Typhoon” (RPA “Typhoon”), St-Petersburg, Russia, which has international accreditation in the Arctic Monitoring and Assessment Program (AMAP) system. Gas-liquid chromatography (GLC) with ECD detection was applied for determination of organochlorine pesticides and PCBs. All samples were analyzed on wet weight (ww) basis.

Prior to being analyzed for POPs, the food samples were thawed, dried (lyophilized), homogenized with weight of NaSO_4_, and extracted. An aliquot of dried homogenized food sample of approximately 2.5 g (equivalent of 10 g wet weight; for blubber 0.5 g ww) was weighed with an accuracy of 0.01 g and placed in a Teflon tube of 100 mL. 500 µL of the keeper (squalane or hexadecane in hexane solution), surrogate standards (*p,p’*-dibromooctafluorobyphenyl (DBOF) and PCB#198), then 5 mL of methanol and 1 mL of acetone were added. After 10–15 min the 30 mL of hexane were added. The mixture was stirred by ultra-sonic extractor for 3–5 min to a suspension form and then was left for 30 min. After the centrifugation the extract was put into a special flask for evaporation. The extraction by hexane was repeated three times with the same volume and the extracts were combined. Insecticides were weighted with an accuracy of 0.01 g and were placed in a 100 mL Teflon tube. Samples of home-brewed alcohol were extracted twice with 30 mL of hexane. The cotton swab with picked-up wash-out was completely transferred to the extraction vessel and then extracted by 30 mL hexane twice.

Concentration of the extract was carried out using a rotary evaporator under vacuum in a water bath (t = 42–44 C), reducing volume of extract to 2 mL. The concentrated extract was transferred from the flask into a test tube by a pipette, and the flask was rinsed with 1 mL of hexane, which was also transferred to a tube. The combined concentrate was dried by adding 1 g of anhydrous sodium sulfate. The purification of the extract was performed by column chromatography on Florisil, then the eluate was concentrated again on a rotary evaporator. The eluate was transferred to a vial by a pipette, the flask was rinsed with 200 μL of hexane, which was also transferred to a vial, and then evaporated to 1 mL by slow flow of high purity nitrogen.

Before injection of the sample into the chromatograph the 50 ng of internal standard (1,2,3,4-tetrachloronaphthalene) was added to each vial. TCN content information was used for volume control of solvent prior to chromatographic analysis. Analysis of samples for organochlorines (OCPs, PCBs) was carried out by the “Kristall 5000.2” device (“Chromatec”, Yoshkar-Ola, Russia). Instrument calibration was performed on the multicomponent standard solution, prepared using individual standards Merck Chemicals GmbH, (Darmstadt, Germany) and Ultra Scientific (North Kingstown, RI, USA). LOD of GLC-ECD method was determined as follows: 0.1 ng/g for samples of food and insecticides; 0.1 ng/L for fermented alcohol; 0.1 ng/sample for wash-outs (corresponds to 2.5 ng/m^2^ or 0.0025 µg/m^2^).

International QA/QC inter-calibration standards for POPs were used under the aegis of the Canadian Northern Contaminants Program. The obtained results were within ±20% of the reference values.

### 2.4. Processing, Analysis and Interpretation of the Data

The analyzed concentrations of POPs in local foods were evaluated in terms of comparisons between different species in coastal Chukotka and in the same species in other circumpolar regions, including the assessment of the temporal trends of POPs. Using food safety regulation standards of pollutants in raw foods, the exceedances over the established limits were determined.

The estimated daily intakes (EDIs) of POPs have been calculated for 60 kg average person body mass (without differentiation for age and gender), based on POPs average concentrations in local foods and on the averaged intake frequencies of each food item, self-reported by the respondents in questionnaire. Using the hypothetically assumed one portion size as 150 g/meal of each foodstuff, the average annual EDIs of pollutants were calculated for each food group. Statistical treatment of the data was carried out using the Microsoft Office-2016 software package.

## 3. Results

### 3.1. Concentrations of POPs in Local Foods

All POPs (Figure 1) show to have very low concentrations in fish, land and marine mammal meat compared to marine mammal blubber (due to lipophilic features of POPs), where they reach: ∑HCH-90, ∑CHL-70, ∑DDT-100, HCB-200, ∑PCB15–150, Mirex-10, Pentachlobenzene-3.7 µg/kg ww.

In the context of the summarized POPs content the gray whale blubber and *mantak* (a Yupik name for the layer for the whale skin with a thin layer of adjacent blubber) is the most contaminated among the studied marine mammal fatty tissues (Figure 2 and Table 2); it is characterized by the highest levels of HCHs, chlordanes and HCB, while DDTs and PCBs are the second greatest (after seal blubber). Whale blubber and mantak contain high levels of b-HCH (up to 70–80 µg/kg), heptachlorepoxide (up to 12), *trans*-chlordane (up to 25), and oxychlordan (up to 5.2), which are very low in all other marine mammals. *trans*-Nonachlor is high only in whale blubber and mantak (21 and 35 µg/kg ww correspondingly) and in seal blubber (up to 30). However whale blubber and mantak does not contain mirex (found in seals, bearded seals and walrus) and pentachlorobenzene (found in seals and bearded seals).

Ringed and spotted seal blubber is the second highest contaminated tissue; levels of DDTs, PCBs and pentachlorobenzene are the highest among compared species of marine mammals. Bearded seal blubber has relatively low levels of HCHs, chlordanes and HCB, but expressed levels of DDTs, PCBs and pentachlorobenzene. Walrus blubber is the least affected among the studied marine mammal blubber; it has the lowest concentrations of all POPs with the exception of mirex, which is surprisingly the highest (7–10 µg/kg). Walrus *kopalkhen* (Chukotkan name of the fermented deboned walrus parts stuffed inside the subcutaneous blubber and aged in subterranean pits, typically over a period of 6 months) generally shows to have half the levels of all POPs, compared to walrus blubber and therefore can be regarded as conditionally uncontaminated food. DDE/DDT ratio is high in all blubber species (range from 7 to 50), which leads to the conclusion that the DDT contamination of all marine mammals is “old”.

### 3.2. Follow-up of Some Legacy POPs in the Local Foods in Coastal Chukotka

Due to the limited number of the collected and analyzed samples the present Chukotka study cannot provide a reliable or statistically significant comparison with the bigger number of samples collected 15 years ago. However, a comparison was carried out with the data on the same species of fish, reindeer and marine mammals collected in the neighboring Chukotsky coastal district in 2001–2002 [15]. Table 3 represents the results of this re-assessment. At the backdrop of the general decrease (or invariability) of the majority of POPs concentrations, the increase tendency of HCB is obvious in almost all marine mammal species (except walrus) and in reindeer. Bearded seal meat demonstrates certain increasing trend of all studied POPs, while bearded seal blubber shows increase in HCB and mirex levels only.

Despite statistical insignificance the increase of mirex levels (in 2016 compared to 2001) in blubber of ringed seal, bearded seal and particularly walrus (Table 3) warrant special attention (whale blubber shows no trace of mirex, same as 15 years ago). Production of mirex was internationally stopped in 1978, and the use of mirex as a pesticide has been banned. Unlike DDTs and HCHs (which were used for fighting botflies in reindeer herding), and PCBs (which were used as frost-resistant additives for the technical fluids in Russian Arctic in 1960s–1980s), mirex was intended for combating ants in southern latitudes, and does not have a known history of use in Russia and the Arctic.

### 3.3. Structure of the Estimated Daily Intakes of POPs

Figure 3 shows the structure of POPs EDIs. More than 90% of all POPs are ingested by local people with marine mammal blubber. Majority of POPs intake is caused by the consumption of whale blubber, including *mantak* (38% of DDTs, 52% of HCHs, 84% of HCB, 34% of PCBs and 61% of chlordanes). The consumption of ringed and spotted seals blubber is responsible for significant share of POPs intake (from 6% of HCB and up to 74% of pentachlorobenezene). Walrus blubber, including *kopalkhen*, contributes substantially less to the intake of POPs: 22% of HCHs and 9% of PCBs, but it contributes 81% of mirex intake. Bearded seal blubber contributes 10% of DDTs, 10% of PCBs, 2% of HCHs, 2% of HCB, 6% of chlordanes, 19% of mirex and 26% of pentachlorobenezene intakes. Marine mammal meat constitutes relatively small portion of organochlorine intake (6–10% of the main POPs). The input from fish is small: 2–5% of HCHs, HCB and chlordanes, and 9–10% of DDTs and PCBs; HCHs in have not been detected in any species of the sampled fish. Terrestrial mammal and fowl meat are negligible for POPs EDIs (Figure 3).

### 3.4. Hygienic Regulations of POPs in Raw Foods

It should be noted that Russian hygienic regulations of legacy POPs in wildlife species [17] are much more comprehensive compared to the international Codex Alimentarius and other foreign standards. Nevertheless the Russian regulations do not cover all the legacy POPs and all the subsistence species, which hampers the evaluation of the “degree” of contamination of different foods by different organochlorines. Russian Allowable Levels (Table 4) of PCBs are established only for the tissues of marine mammals and fish; chlordanes are regulated only in meat of terrestrial mammals and birds; DDTs and HCHs are not regulated in fish; HCB is not regulated at all. Foreign standards (e.g., Codex Alimentarius) are even less rigorous in respect to Maximum Residue Levels (MRLs) of POPs in food, and do not fill the shortcomings of the Russian regulations.

### 3.5. Exceedances of the Analyzed Concentrations of Pollutants in Foods over the Russian Allowable Levels

The highest concentrations of POPs in all analyzed samples of land and marine mammals (meat and blubber), goose and fish (freshwater, migratory and marine) do not exceed the Russian Allowable Levels (AL); only the ringed seal blubber has 100 µg/kg ww of ∑DDTs (equal to AL). High concentrations of HCB in whale blubber and mantak (180–200 µg/kg ww) could not be compared to not existing AL of HCB in foods.

### 3.6. Additional Sources of in-Home POPs Contamination of Food

Samples of *braga* (home-brewed low-alcohol drink, usually manufactured in plastic containers), domestic insecticides (gels in tubes) and wash-outs of kitchen walls which were collected in Enmelen settlement, did not contain chlordanes, mirex or tetra/penta/hexachlorobenzenes. Concentrations of HCHs, DDTs and PCBs in those samples are presented in Table 5.

Braga is a source of additional direct exposure: by consuming 10 L of braga per month a person may be ingesting up to 0.25 µg of ∑HCHs, 0.3 µg of ∑DDTs and 1 µg of ∑PCBs. These amounts are much less than the exposure to the same pollutants one gets from consuming fatty marine mammals, but they are comparable to the exposure resulting from 1–2 kg of land mammals or certain species of marine fish. Moreover, these animals are free of HCHs (the levels are below LOD), while braga contains relatively high HCHs concentrations; all three isomers of HCH (α, β and γ) are present in braga (Table 5).

Levels of all three groups of POPs in the two samples of insecticides (both of Russian production) and wash-outs are significantly lower than that in braga. It is clear that the wash-outs show the multi-year application of insecticides on kitchen walls. Concentrations of HCHs and DDTs are similar in both matrices, and one sample of wash-outs shows even higher level of DDTs. Concentrations of PCBs are higher in the insecticides than in wash-outs. Generally, the POPs contamination of walls in the kitchens in coastal Chukotka in 2016 and in 2001 [15] are comparable, and demonstrate that in-home POPs have had the same presence in the dwellings of coastal Chukotka during all years passed. Due to the variability of the insecticides sampled in 2001 in Uelen (with certain types being nearly free of POPs and others having high levels), it is impossible to draw comparison with the insecticides sampled in Enmelen.

Regarding DDTs from non-food sources, it should be noted that DDE/DDT ratio (which indicates the oldness of DDTs occasion) varies among the different samples of braga, insecticides, and wash-outs; no consistent pattern is visible. One braga sample and one wash-out sample show the predominance of “fresh” *p,p’*-DDT (while *p,p’*-DDE is not detectable or low); one braga sample demonstrates equal presence of both metabolites; other samples contain either low levels of DDT metabolites or show the prevalence of “old” *p,p’*-DDE.

PCBs are presented in all samples of braga, insecticides and wash-outs exclusively by the following seven congeners: #52, #99, #101, #105, #118, #138, #153 (levels of other eight congeners were below the LOD).

## 4. Discussion

### 4.1. Diet of Marine Mammals

Section 3.1 demonstrates the significant differences of various POPs levels in blubber of the studied marine mammal species. Can variations in the diet of individual species account for these differences? We know that the ringed and spotted seals are typical fish eaters [18]. To the contrary, walrus, bearded seal and gray whale are typical benthic feeders having similar diet; they all eat different kinds of benthic invertebrates, including worms, gastropods, cephalopods, crustaceans (shrimps, crabs, amphipods), mollusks (clams and whelks), tunicates, sea cucumbers, etc. The slight differences are: walrus prefers bivalve mollusks, bearded seal prefers shrimps and crabs, gray whale prefers shrimp-like amphipods.

POPs levels in the analyzed species of fish in the present study are very low (total amount up to 3–7 µg/kg ww). Information on contaminants bioaccumulation in various species of benthic invertebrates in the Arctic is insufficient, but generally the POPs levels in these organisms (not subject to biomagnification) are less than in fish. It is clear that relatively small ringed and spotted seals eat less weight of food (in this case fish) than the big walrus, bearded seal, and especially whale (an adult gray whale eats up to 2 tons of invertebrates daily) [18]. Assuming that all marine mammals eat daily approximately the same amount of food in proportion to body mass, they should accumulate similar doses of POPs. In this context it is hard to explain the variability in the analyzed levels of POPs in different species of marine mammals, which feed in the same place. Considering the results of the analyses of the samples of walrus, bearded seal and gray whale (all eat invertebrates) delivered in the present study (Section 3.1), several questions arise: why the gray whale blubber, highly contaminated by most of POPs (particularly by chlordanes), does not contain mirex and pentachlorobenzene? Why the walrus blubber is almost free of chlordanes and DDTs, but contains β-HCH, PCBs and the highest level of mirex? Why the bearded seal blubber has very low concentrations of ∑HCH, etc.? The logic suggests that the differences in POPs contamination of the studied marine mammal species could be attributed to the diversity of places where they feed, where the contamination of invertebrates is unequal.

### 4.2. Geographic Comparisons

Given the shared resource base and continuities in cultures and cuisines, it is helpful to carry out geographic comparison of contaminants in marine mammals for the Bering Strait region as a whole. Due to the limited number of publications over the past decade, we also include data comparisons with the older publications.

*Walrus*. The only recent data available on Pacific walrus is from the samples collected in 2012–2016 in Saint Lawrence Island (about 100 km to the south-east from the Chukotka coast). A comparison with the present Chukotka walrus blubber results with that study’s results from Saint Lawrence Island [19] demonstrates close similarity of mean concentrations of ∑HCHs (55.0 vs. 70.3 µg/kg ww), β-HCH (53.6 vs. 65.9 µg/kg), ∑DDTs (2.7 vs. 4.7 µg/kg), HCB (0.88 vs. 0.27 µg/kg), ∑PCBs (40.5 vs. 38.6 µg/kg) and mirex (8.6 vs. 5.1 µg/kg) in walrus blubber (Table 4); the exception are the chlordanes levels: ∑CHLs (2.5 vs. 38.7 µg/kg ww) mainly due to oxychlordane in the Saint Lawrence Island sample (33 µg/kg ww). ΣHCH in blubber of Saint Lawrence Island walruses had lower concentrations than Pacific walruses sampled in 1993–1996 [20] and lower than the Atlantic walruses in four locations of the Canadian Arctic [21].

*Ringed, spotted and bearded seals*. There is one publication with POPs data on the ringed seal blubber [22] sampled in 2005 in Vankarem (Chukchi Sea coast of Chukotka): levels of ∑HCHs, ∑CHLs, ∑DDTs and ∑PCBs are closely similar with the corresponding levels of the present Chukotka study. However in Vankarem the levels of α-HCH and oxiclordane are much higher, while the levels of HCB, mirex and pentachlorobenzene are much lower than in the samples in Enmelen, collected in the present study.

Levels of legacy POPs in the blubber of ringed seals (samples collected in 2003–2010) have been studied [13] in 12 Canadian Arctic communities (Sachs Harbour, Ulukhaktok, Arctic Bay, Gjoa Have, Grise Fjord, Resolute, Qikiqtarjuaq, Pangnirtung, Arviat, Inukjuak, Ungava, Nain) with the range of geometric mean concentrations: ∑HCHs (51–148 µg/kg lw), ∑CHLs (101–283 µg/kg lw), ∑DDTs (107–600 µg/kg lw), ∑CBz (18–52 µg/kg lw), ∑PCBs (247–912 µg/kg lw). Present Chukotka ringed seal blubber concentrations of ∑HCHs, ∑DDTs and ∑CBz are similar to the corresponding lowest POPs levels in the Canadian study, while the Chukotka levels of ∑CHLs and ∑PCBs are 2–3 times lower than the corresponding lowest levels (Table 2).

Serious discrepancies in levels and structure of ∑HCH in ringed seal blubber should be noted. In the blubber samples from all of Chukotka’s marine mammal species analyzed in the present study (ringed, spotted and bearded seals, walrus and gray whale) the β-HCH is dominating (70–90% of ∑HCH); in the ringed seal blubber the β-HCH portion is 81% (Table 2). On the contrary in the Canadian study the ringed seal blubber ∑HCH consisted mainly of α-HCH, which was represented by 57–79% (in 11 settlements, except Ungava, where this index was 36%) of the total sum of HCH isomers [13]. In the Vankarem study [22] α-HCH in the ringed seal blubber constituted about 50%.

An Alaska-based publication [19] provides the mean levels of POPs in the blubber of ringed, spotted and bearded seals sampled in Alaska (Bering and Chukchi seas) in 2003–2007, which were generally much higher than the corresponding levels in the blubber of ringed, spotted and bearded seals analyzed in the present Chukotka study: ∑CHLs is about 3; 20 and 10 times higher correspondingly, ∑DDTs is about 1.5; 5 and 2.5 times higher, ∑PCBs is about 2; 7 and 4 times higher correspondingly. Only the levels of ∑HCHs in ringed seals blubber were similar in both studies (about 50 µg/kg ww), while the levels of ∑HCHs in spotted and bearded seals blubber were about 5 and 2 times higher, correspondingly, in the Alaska-based study.

Concentrations of ∑HCHs, ∑CHLs, ∑DDTs and ∑PCBs (HCB not presented) in ringed and bearded seals blubber sampled in Barrow, Alaska in 1997–1999 [23], were even higher than Alaskan levels of 2003–2007 [19]. Nevertheless the HCB levels in ringed and bearded seals blubber from Barrow were very close to the respective levels in the present Chukotka study: 17.3 vs. 19.9 µg/kg ww in ringed seals, and 6.7 vs. 8.1 in bearded seals, correspondingly.

*Gray and bowhead whales*. Levels of ∑CHLs and ∑DDTs (∑HCHs not presented) in gray whales blubber collected during a Russian subsistence harvest in 1994 in the Bering Sea, Chukotka [24] were 2–2.5 times higher than in the present Chukotka study; ∑PCBs levels were 6 times higher, but HCB concentrations were very similar (about 200 µg/kg ww).

Concentrations of ∑HCHs, ∑CHLs, ∑DDTs and ∑PCBs in bowhead whales blubber sampled in Barrow, Alaska, in 1997–1999 [25] were 3–6 times higher, but the HCB levels were almost twice lower (184 vs. 100 µg/kg ww) than in the present Chukotka study. Levels of all investigated legacy POPs in whale *mantak* (called *maktak* in the Inupiaq language spoken in Barrow) in both studies were lower than in whale blubber: averagely 1.5–2 times in the present Chukotka study, and up to 2–4 times in the Barrow study.

Thus the legacy POPs in marine mammals (both pinnipeds and cetaceans) sampled in Chukotka were generally much lower than in the species sampled in Alaska and northern Canada. This phenomenon warrants further investigation with a consideration of animal migration routes.

### 4.3. Annual Migration of Marine Mammals in the Bering-Chukchi-Beaufort Seas

Despite the demonstrated success of the use of satellite/radio transmitters and dive recorders for tracking marine mammals migration routes, there is still no clear understanding of habitat partitioning of different species in the northern Pacific and adjacent Arctic in different seasons (particularly regarding male-female, adult-young animals). Observations in Russian waters inform that the ringed, spotted and bearded seals from the Bering Sea migrate (starting in late spring and early summer) to the Chukchi Sea (moving through the Bering Strait) up to Wrangel and Herald Islands, and in autumn–back southward [26]. International study on walruses allowed determining 2 main routes of their spring migration from the Bering Sea to Chukchi Sea: western–to Wrangel Island, and eastern–to Alaska northwestern coast. Generally two populations of walruses remain separately in Russian and US waters through September, and then in October both populations move back towards Bering Sea where they can mix. There have been observations that some adult male walruses do not migrate to the north from the Bering Sea [27].

Observations in Alaskan Bering Sea waters demonstrate that the majority of spotted and bearded seals also move northwards into the Chukchi and Beaufort seas in the late spring and early summer as the ice edge recedes, and in autumn they migrate southwards through the Bering Strait into the Bering Sea as the pack ice advances [28,29,30]. Sub-adult ringed seals tagged in Kotzebue Sound in October traveled south from the Chukchi Sea into the Bering Sea as sea ice coverage increased during November and December, remained ~1000 km south near the ice edge during winter and returned north in the spring with the receding ice edge. Adult ringed seals remained in the Chukchi and northern Bering seas, where their movements were more localized [31]. Some of the ringed seals tracked from the eastern Beaufort Sea in September, demonstrated their winter locations on the Chukotka northern coast, but some of the tracked seals crossed the Bering Strait to spend the winter in Bering Sea [32].

Gray whales from the Bering and Chukchi seas migrate annually, starting in October, to California Peninsula and then in spring coming back (up to 20 thousand km round way) to the Arctic [33]. Bowhead whales from the Bering Sea in early spring migrate through the Chukchi Sea passing Point Barrow in late spring through the Beaufort Sea to Amundsen Gulf, Canada. In summer they move across the entire Beaufort Sea leaving Point Barrow in the fall and spend weeks along the northern Chukotka coast before entering the Bering Sea. Winter movements of the bowhead whales are concentrated in the western Bering Sea from Bering Strait to the ice edge [34].

Thus the migration routes of the pinnipeds and cetaceans in the Bering-Chukchi-Beaufort seas are very complicated, altering and depending on climate change when the sea ice areas (and therefore the feeding opportunities) vary substantially during the same seasons from year to year [27]. It can easily happen that when a sampling of a seal take place in Chukotka waters, there is no guarantee that this seal is local because it could migrate to Chukotka from another place (e.g., from Beaufort Sea) where its population resides and where this seal is feeding for a good part of the year.

### 4.4. Marine Currents and River Estuaries of the Bering-Chukchi-Beaufort Seas

It appears that the most important aspects of the above problem are the local sources of contaminants and pathways of their transport in connection with the peculiarities of habitats and migration of the marine mammals. Let us look at the marine currents and river estuaries of the Bering-Chukchi-Beaufort seas [35,36,37]. Alaska Peninsula, together with the Aleutian Archipelago, forms the natural intercontinental “dam” fencing the Bering Sea and protecting it from the Pacific currents. Bering Sea is divided almost equally between a deep Aleutian basin (southern part) and the Bering Sea shelf (northern part). Northeastern part of the Bering Sea shelf is very shallow (about 50 m) which is similar to the shallow Chukchi Sea. There are only two major currents in the Bering Sea which rush to the Bering Strait: (a) powerful Alaska Coastal Current, which debouches through the Unimak Pass at the end of the Alaska Peninsula and goes along the Alaskan coast, and (b) Bering Sea Current, which debouches from Alaska Stream through the passes in the Aleutian Archipelago and goes along the watershed of the Aleutian basin and Bering Sea shelf towards Anadyr bay and Chukotka Peninsula coast (Figure 4).

After passing the Bering Strait the Bering Sea Current goes into the central part of Chukchi Sea while the Alaska Coastal Current goes to the Beaufort Sea along the coast of Alaska. All these explanations are needed to substantiate the idea that the rivers inflows to Bering Sea from Alaska are captured exclusively by the Alaska Coastal Current and then are moving along the coast. Contaminants in discharges of the Yukon River (collected from the catchment area of 0.8 mln km^2^) do not dissolve in the Bering Sea waters, but are transported by Alaska Coastal Current directly to the Beaufort Sea, where the amount of contaminants further increases significantly by the discharges of the great Mackenzie River (catchment area 1.8 mln km^2^).

On the Russian Chukotka side the only relatively large river inflowing the Bering Sea is Anadyr River (catchment area 0.19 mln km^2^), which is four times smaller than the Yukon River. Our hypothesis is as follows: due to the river inflows and specific configuration of the marine currents in the Bering Sea, the feed base (benthic invertebrates) for marine mammals in Chukotka coastal waters of western Bering Sea and western and central Chukchi Sea is less contaminated by POPs and other pollutants compared to that of the Alaskan coastal waters of eastern Chukchi Sea and particularly southern Beaufort Sea. Perhaps that is why the levels of POPs in marine mammals sampled in the nearby Chukotka are much lower than the corresponding levels in the animals sampled near Alaska or northwestern Canada.

The influence of big river discharges on the contamination of ringed seals was distinctly demonstrated in studies in the Russian Arctic [22]. Concentrations of POPs in the ringed seal blubber sampled in 2001–2002 in the estuaries of rivers Severnaya Dvina (catchment area 0.36 mln km^2^), Pechora (0.32 mln km^2^) and Yenisey (2.58 mln km^2^) were up to 10–70 times higher than the corresponding levels in ringed seal blubber sampled near Chukotka in 2005 [22] or in the present study. The following regularity is observed: the more powerful the river, the more contaminants in seal blubber; for example the maximal concentrations were (in the sequence from west to east: Severnaya Dvina-Pechora-Yenisey-Chukotka)–∑CHL: 95—216—897—79 µg/kg ww; ∑DDT: 1360—1920—7130—119 µg/kg ww; ∑PCB: 2420—1490—3940—238 µg/kg ww; mirex: <0.05—6.3—19.2—1.5 µg/kg ww. It is interesting that the mirex levels in Chukotka’s Vankarem (northern coast of peninsula) were lower than 5.28 µg/kg ww in Enmelen (sampled in the present study), located in the Anadyr Bay. Also it should be noted that ∑HCH and HCB were relatively similar for all 4 regions, which may indicate that river transportation of these contaminants is not a significant factor.

### 4.5. Additional Sources of Food Contamination

Questionnaire of the Indigenous people in Enmelen has identified that home-brewed alcohol is consumed by 19% of respondents; 14.3% of the respondents use insecticides at home to combat cockroaches. The levels of HCHs, DDTs and PCBs found in the 2016 samples of braga raise some questions as to their origins. Much higher concentrations of these pollutants were found in the 2003 samples in Uelen [15,16], but their continued presence at relatively high levels 15 years later can no longer be attributed to POPs-contaminated containers (if we are to assume that some POPs could be stored in these containers) used to make and store the braga—an explanation suggested earlier. We know that local braga manufacturers use their ware continuously over many years, which should be effective in rinsing out the POPs. The fact that we have found three different groups of POPs present in braga stored in a single container further indicates that the initial contents of the container are not likely to be the source, since all three POPs groups could not mix together in the pre-braga environment of the wares. Another possible explanation is the source of the sugar used to make braga (corns, peas or other carbohydrates) for the fermentation, which may have been contaminated by several POPs simultaneously in the place of cultivation of raw matters or the subsequent processing (e.g., in China where POPs are still in active use); we do not have further insight to explore this possibility. Finally, and, in our opinion, most likely, the explanation could be in the chemistry of the fermentation processes. Interaction of plastic with alcohol (also with salts, acids and other products formed during the processes of fermentation or food pickling) triggers the elution of various organic substances from the plastic. These substances could in turn cause unforeseen chemical reactions, which could result in the formation of new toxic compounds, if the “precursors” and “catalysts” are involved in chemical reactions. These processes, as well as chemical composition of the plastic containers which are in use, demand further investigation. For now it is necessary to conduct outreach and urge people to exclude the use of plastic containers for pickling, aging, and other ways of preserving food products and for the preparation of alcoholic beverages. For such purposes, chemically inert glass, steel, enamel, or wooden containers should be used.

## 5. Conclusions

To our knowledge this is the only comprehensive study on POPs in local subsistence foods from the coastal Chukotka since the beginning of 2000s.

Concentrations of the studied legacy POPs were very low in fish and the meat of land and sea mammals and fowl. The levels of POPs in the blubber of marine mammals were relatively high (up to 100–200 µg/kg ww) but not exceeding the allowable limits. Gray whale blubber and whale *mantak* were the most contaminated food, followed by the ringed, spotted and bearded seal blubber, and then by walrus blubber and walrus *kopalkhen* (fermented blubber).

Structure of the estimated daily intakes (EDIs) of legacy POPs has demonstrated that more than 90% of the dietary exposure to POPs of local people is due to the consumption of marine mammals blubber.

Temporal comparisons of POPs in local foods in the present study and the previous coastal Chukotka study (15 years ago) has shown that, at the backdrop of general decrease (or invariability) of the majority of POPs, an increasing tendency of HCB, mainly in marine mammals, could be noted. These observations are generally consistent with the circumpolar and global declining trends of the majority of legacy POPs in marine, freshwater, and terrestrial biota during recent decades.

Geographic comparisons have shown that the legacy POPs in marine mammals (both pinnipeds and cetaceans) sampled in Chukotka were generally much lower than in the corresponding species sampled in Alaska and northern Canada. We suppose that due to the transportation of the discharges from the Yukon River by the Alaska Coastal Current from the Bering Sea and along the Alaska coast to the Beaufort Sea (where the amount of contaminants heavily increase by the discharges from the Mackenzie River), the feed base for marine mammals in Chukotka coastal waters is less contaminated by POPs (and other pollutants) compared to Alaskan and northern Canada coastal waters.

Analysis of the additional sources of in-home food contamination (home-brewed alcohol, domestic insecticides, and wash-outs from the kitchen walls) has revealed relatively high levels of HCHs, DDTs and PCBs, which were lower than that in coastal Chukotka 15 years ago, but still represent a share of the dietary exposure of local people to POPs.

## Figures and Tables

**Figure 1 ijerph-16-00695-f001:**
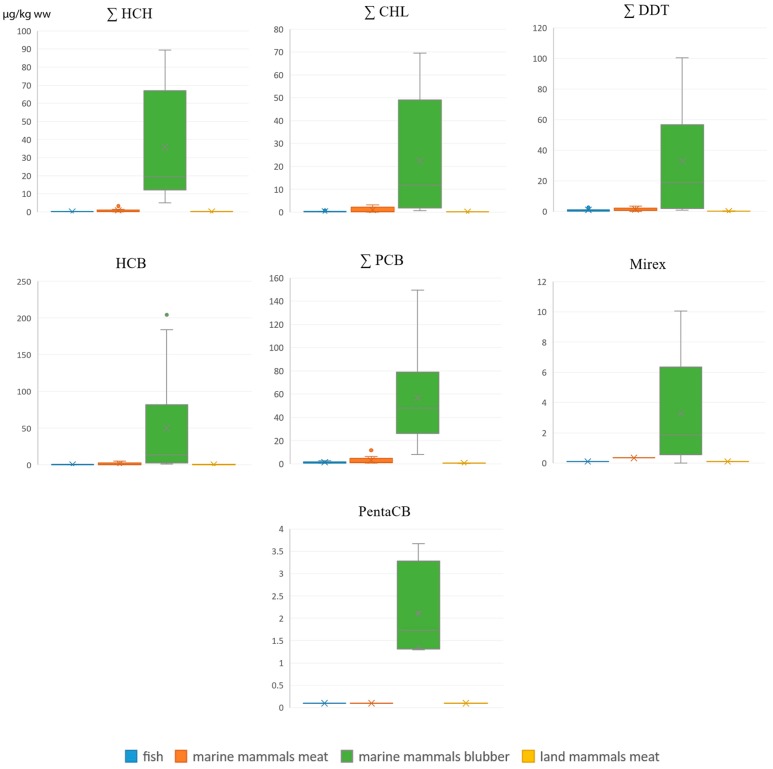
Concentrations of POPs in fish, marine mammal meat and blubber and in land mammals meat, µg/kg ww.

**Figure 2 ijerph-16-00695-f002:**
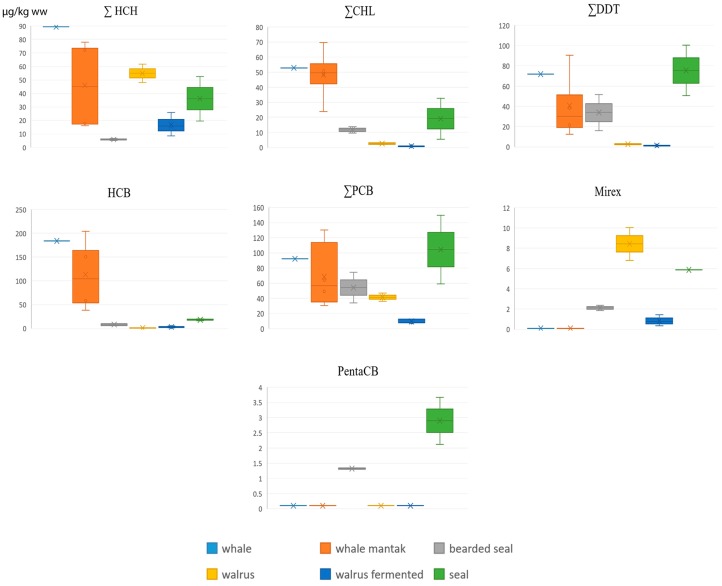
Concentrations of POPs in blubber of marine mammal species, µg/kg ww.

**Figure 3 ijerph-16-00695-f003:**
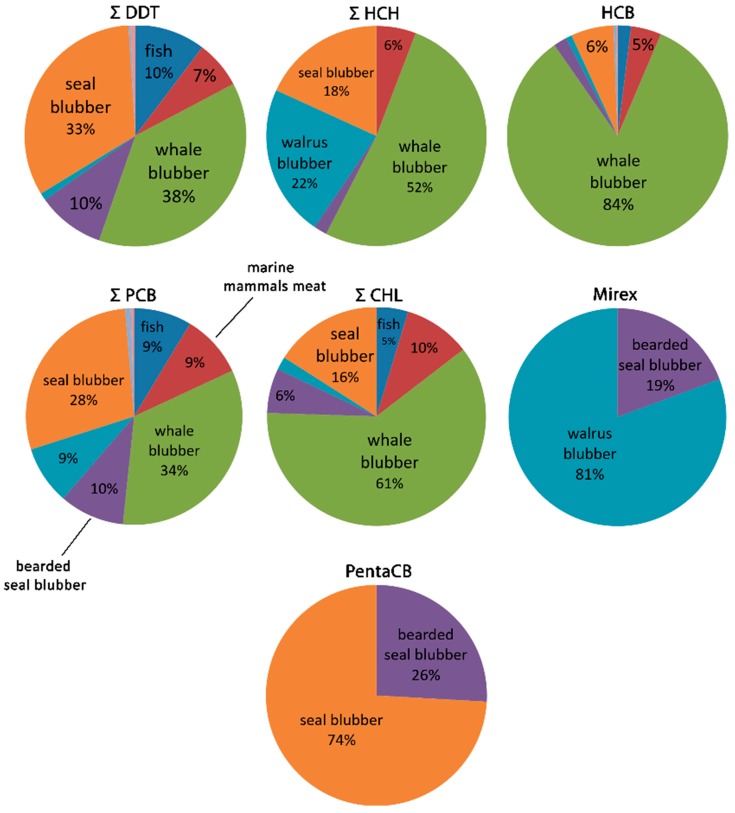
Structure of the Estimated Daily Intakes (EDIs) of POPs by local food consumption, %.

**Figure 4 ijerph-16-00695-f004:**
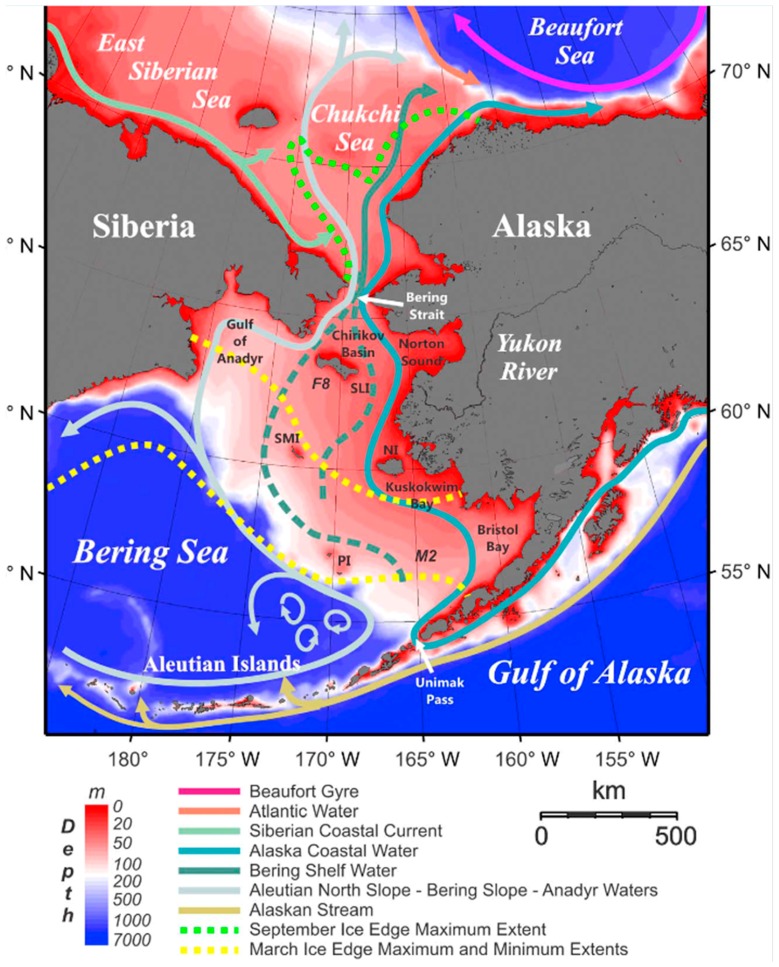
Scheme of the major marine currents in the Bering, Chukchi and Beaufort Seas (Figure is cited from Danielson et al., 2011 [35]).

**Table 1 ijerph-16-00695-t001:** Collected and analyzed samples of local foods and indoor matters (numbers and locations).

Species	Tissue	*n* of Single Samples	*n* of Pools	Location
**Fish**		**11**	**5**	
Arctic char (*Salvelinus alpinus*)	muscles	-	1	Achon lake
Arctic char (*Salvelinus alpinus*)	muscles	3	-	Enmelen
Grayling (*Thymallus thymallus*)	muscles	1	-	Enmelen
Humpback salmon (*Oncorhynchus gorbuscha*)	muscles	-	1	Nunligran
Chum salmon (*Oncorhynchus keta*)	muscles	1	-	Sireniki
Coho salmon (*Oncorhynchus kisutch*)	muscles	1	-	Enmelen
Sockeye salmon (*Oncorhynchus nerka*)	muscles	1	-	Nunligran
Pollack (*Pollachius pollachius*)	muscles	2	-	Enmelen
Pollack (*Pollachius pollachius*)	muscles	1	-	Sireniki
Cod (*Gadus morhua*)	muscles	-	1	Sireniki
Starry Flounder (*Platichthys stellatus*)	muscles	-	1	Nunligran
Saffron cod (*Eleginus gracilis*)	muscles	-	1	Nunligran
Saffron cod (*Eleginus gracilis*)	muscles	1	-	Enmelen
**Marine Mammals**		**28**	**-**	
Gray whale (*Eschrichtius robustus*)	meat	2	-	Enmelen
	meat	1	-	Nunligran
	blubber	1	-	Nunligran
	mantak	2	-	Enmelen
	mantak	1	-	Nunligran
	mantak	1	-	Sireniki
Pacific walrus (*Odobenus rosmarus*)	meat	2	-	Sireniki
	meat	1	-	Nunligran
	meat	1	-	Enmelen
	blubber	2	-	Enmelen
	kopalkhen	4	-	Enmelen
Bearded seal (*Erignathus barbatus*)	meat	2	-	Sireniki
	meat	1	-	Nunligran
	meat	1	-	Enmelen
	blubber	2	-	Enmelen
Ringed seal (*Phoca hispida*)	meat	1	-	Sireniki
	blubber	1	-	Enmelen
Larga seal (*Phoca largha*)	meat	1	-	Sireniki
	blubber	1	-	Sireniki
**Terrestrial mammals**		**6**	**-**	
Reindeer (*Rangifer tarandus*)	meat	2	-	Enmelen
	meat	1	-	Sireniki
Arctic hare (*Lepus arcticus*)	meat	1	-	Enmelen
	meat	1	-	Nunligran
	meat	1	-	Achon lake
**Birds**		**1**	**-**	
Snow goose (*Anser caerulescens*)	meat	1	-	Sireniki
**Indoor matters**		**4**	**2**	
*Braga* (fermented alcohol)		-	2	Enmelen
Insecticides (gel)		2	-	Enmelen
Wash-outs (from domestic kitchen walls)		2	-	Enmelen
**Total: 57**		**50**	**7**	

**Table 2 ijerph-16-00695-t002:** POP concentrations in marine mammal blubber, collected in Chukotka Providensky district in April 2016. Levels, average levels (range), ng/g ww.

	Ringed Seal	Spotted Seal	Bearded Seal	Walrus	Walrus	Gray Whale	Gray Whale
	Blubber	Blubber	Blubber	Blubber	*Kopalkhen*	Blubber	*Mantak*
*n* samples	1	1	2	2	3	1	4
α-HCH	8.67	5.25	1.77–2.02	1.21–1.24	0.64 (0.36–0.89)	7.71	4.0 (1.6–6.73)
β-HCH	43.12	14.35	2.86–4.88	46.61–60.52	16.06 (7.92–24.7)	79.72	40.31 (13.87–68.76)
γ-HCH	0.86	<0.1	0.17–0.18	<0.1–0.36	<0.1–0.25	1.91	1.68 (0.64–2.74)
∑HCH	52.65	19.60	5.1–6.8	48.2–61.8	16.8 (8.6–25.8)	89.34	46.0 (16.1–77.9)
heptachlor	<0.1	<0.1	0.55–0.62	<0.1–0.71	<0.1–0.52	<0.1	<0.1–0.42
heptachloroepoxide	<0.1	<0.1	2.54–3.16	<0.1–0.58	<0.1	11.99	6.52 (2.92–9.72)
t-chlordane	<0.1	<0.1	0.12–0.2	0.7–1.01	0.23 (0.17–0.29)	15.13	18.91 (12.3–25.25)
oxychlordane	<0.1	<0.1	<0.1–2.19	<0.1	<0.1	3.27	2.42 (0.95–5.21)
c-nonachlor	<0.1	<0.1	<0.1	<0.1	<0.1	<0.1	<0.1
c-chlordane	2.54	<0.1	0.48–0.85	<0.1	<0.1	1.54	1.06 (0.68–1.76)
t-nonachlor	30.14	5.56	5.21–7.55	0.86–1.39	0.57 (0.51–0.67)	20.94	19.72 (7.09–34.93)
∑Chlordanes	32.68	5.56	9.67–13.8	2.14–3.11	1.32 (1.2–1.48)	52.87	48.2 (24.0–69.6)
*o,p’*-DDE	1.59	0.86	0.21–0.32	<0.1–0.88	0.24 (0.23–0.24)	2.21	1.6 (0.44–3.03)
*p,p’*-DDE	96.62	47.06	15.18–49.2	1.8–2.16	0.63 (0.39–0.82)	48.71	25.71 (8.23–54.57)
*o,p’*-DDD	<0.1	<0.1	<0.1	<0.1	<0.1	<0.1	<0.1–4.55
*p,p’*-DDD	<0.1	0.67	<0.1–0.14	<0.1	<0.1	13.47	6.56 (1.4–14.78)
*o,p’*-DDT	<0.1	<0.1	<0.1–0.16	<0.1	<0.1	<0.1	<0.1
*p,p’*-DDT	2.16	1.90	0.7–1.78	<0.1–0.51	0.43 (0.35–0.58)	7.36	5.68 (2.35–13.5)
∑DDT	100.37	50.49	16.1–51.6	2.31–3.04	1.3 (0.97–1.64)	71.75	40.7 (12.4–90.4)
Mirex	5.87	<0.1	1.86–2.38	6.81–10.05	0.85 (0.33–1.45)	<0.1	<0.1
Pentachlorobenzene	2.12	3.67	1.3–1.35	<0.1	<0.1	<0.1	<0.1
HCB	19.96	16.30	5.95–10.24	0.79–0.97	2.97 (1.1–4.87)	183.70	112.9 (38.45–204.2)
PCB-118	6.78	8.02	3.38–12.46	3.9–5.39	1.86 (1.64–2.12)	9.53	8.16 (2.66–19.29)
PCB-138	28.08	8.75	5.97–15.22	2.78–4.15	0.7 (0.53–0.89)	11.85	6.77 (2.46–14.14)
PCB-153	52.97	17.24	10.07–22.42	22.13–30.55	4.22 (2.67–6.31)	16.23	8.38 (2.52–18.21)
∑PCB15	149.60	59.02	34.2–74.3	36.2–46.8	9.7 (7.9–12.8)	92.20	68.5 (30.5–130.1)

**Table 3 ijerph-16-00695-t003:** Temporal comparisons (2016 vs. 2001) of some POPs concentrations in local subsistent foods in coastal Chukotka.

		∑HCH	∑CHL	∑DDT	HCB	∑PCB	Mirex
Arctic char (freshwater)	muscles	↓	↓	↓	→	↓	→
Flounder (marine)	muscles	↓	→	→	→	↓	→
Reindeer	muscles	↓	→	↓	↑	↓	→
Ringed seal	muscles	↓	↓	↓	→	↓	→
Spotted seal	muscles	↓	↓	↓	→	↓	→
Bearded seal	muscles	↑	↑	↑	↑	↑	↑
Walrus	muscles	↓	→	→	→	→	→
Gray whale	muscles	→	→	→	↑	→	→
Ringed seal	blubber	↓	↓	→	↑	→	↑
Spotted seal	blubber	↓	↓	→	↑	→	→
Bearded seal	blubber	→	↓	→	↑	→	↑
Walrus	blubber	↓	→	↓	↓	→	↑
Gray whale	blubber	→	→	→	→	→	→

Green arrows down = decrease; black arrows flat = no change; red arrows up = increase.

**Table 4 ijerph-16-00695-t004:** Russian Allowable Levels of POPs in raw foods, µg/kg ww [17].

	∑DDTs	∑HCHs	∑PCBs	HCB	∑CHL
Meat of land mammals	100	100	ne	ne	50
Meat of marine mammals	200	200	2000	ne	ne
Blubber of marine mammals	100	200	3000	ne	ne
Birds (muscles and eggs)	100	100	ne	ne	500
Fish (muscles)	ne	ne	2000	ne	ne
Freshwater fish (muscles)	300	30	ne	ne	ne
Marine fish (muscles)	200	200	ne	ne	ne
Fish caviar	2000	200	2000	ne	ne
Fish liver	3000	1000	5000	ne	ne

ne—not established.

**Table 5 ijerph-16-00695-t005:** Concentrations of HCHs, DDTs and PCBs in samples of home-brewed alcohol, insecticides and wash-outs of kitchen walls in Enmelen settlement.

	Braga		Insecticides		Wash-Outs	
	(Home-Brewed Alcohol)		(Gels in Tubes)		(Kitchen Walls)	
	ng/L		ng/mL		ng/filter	
	Container 1	Container 2	Gel 1	Gel 2	Flat 1	Flat 2
a-HCH	5.39	1.6	nd	nd	nd	nd
b-HCH	11.43	3.17	0.14	0.38	0.16	0.11
g-HCH	6.58	7.95	nd	nd	0.1	nd
∑HCHs	23.4	12.72	0.14	0.38	0.26	0.11
4.4 DDE	nd	1.5	nd	1.1	1.18	0.9
4.4 DDT	29.12	1.23	nd	nd	4.85	nd
∑DDTs	29.12	3.51	0.1	4.71	7.53	0.9
∑PCB15	95.78	28.88	1.3	4.28	0.83	0.27

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
