# Peer review of "Traditional Diet and Environmental Contaminants in Coastal Chukotka II: Legacy POPs"

_ijerph, 2019, doi:10.3390/ijerph16050695_

Round 1

Reviewer 1 Report

This is a descriptive study on different POPs in Chukotka region. As such it may be of interest although many analyses suffer from small numbers and it is hard to see how representative they are. The introduction is quite long and the authors may consider, how much of the material belongs to a review rather than an individual study.

Some details

Overall: replace mcg with μg

lines 206-207: does the 10 g sample apply also for blubber?

line 208: mkl??

line 209: DBOF? spell out

line 221: What was the eluent used with Florisil column? Was fat extracted retained in Florisil or was part of it separated to a prefraction before POPs?

line 225: 1,2,3,4-TCN? spell out

line 227: organochlorines

line 345: absolutely free of HCHs?? Do you mean, not detectable with this analysis; below LOD?

Line 364: present; again, this is dependent on your detection limit

Lines 391-392: did you consider possible different kinetics and half-lives in different species?

Author Response

Comments and Suggestions for Authors

·         This is a descriptive study on different POPs in Chukotka region. As such it may be of interest although many analyses suffer from small numbers and it is hard to see how representative they are.

Answer: Our study settlements are remote communities that are difficult to reach due to extremely limited transportation options that are also vulnerable to weather. We worked within the logistical and budgetary constraints to collect samples of the key species (as inclusive and diverse as possible) of local fauna and flora to assess the present day contamination of the maximum wide spectrum of species, which are consumed by local people, being the main sources of POPs exposure, and to compare these detected values with previous studies in the region and in neighboring foreign Arctic territories. Under such circumstances each sample (collected and analyzed) is very important for the entire goal and different tasks of the project.

·         The introduction is quite long and the authors may consider how much of the material belongs to a review rather than an individual study.

Answer: 1) We believe our Introduction section is of reasonable length in proportion to the rest of the article  2) The present follow-up study of the POPs legacy in coastal Chukotka draws substantial, and often very strong interest among environmental health researchers, especially those working in Circumpolar Arctic because coastal Chukotka is characterized by the highest (compared to other Russian Arctic regions) levels of POPs contamination found in local foods and the highest POPs exposure levels for local indigenous people (based on the results of the 2001-2003 study). We intended for the background overview provided in the Introduction section to be sufficiently broad and effective in summarizing the relevant data (incl. trends of environmental POPs in circumpolar Arctic). We believe it is important to offer readers an overview that strives to be comprehensive in its coverage of the existing knowledge in our study area against which they are to consider the results of the present follow-up study in Chukotka.

In order to improve the flow we divided the section Introduction into three subsections: 1.1. Overview; 1.2. Circumpolar POPs trends in local foods, a brief summary; 1.3. Objectives and tasks of the follow-up study in coastal Chukotka.

Some details

·         Overall: replace mcg with μg

Answer: Done in the text.

·         lines 206-207: does the 10 g sample apply also for blubber?

Answer: for blubber we used 1g sample. Added in the text.

·         line 208: mkl?

Answer: corrected to µL.

·         line 209: DBOF? spell out

Answer: 4,4’-dibromo-octafluorobyphenyl. Added in the text.

·         line 221: What was the eluent used with Florisil column? Was fat extracted retained in Florisil or was part of it separated to a prefraction before POPs?

Answer: The eluent was successively 30 ml of hexane, then 40 ml of hexane and dichloromethane mixture (1:1). We did not use preliminary separation of fat; it was separated on the Florisil column (capacity of the fat separation was near 0.5 grams per sample).

·         line 225: 1,2,3,4-TCN? spell out

Answer: 1,2,3,4- tetrachloronaphthalene. Added in the text.

·         line 227: organochlorines

Answer: corrected.

·         line 345: absolutely free of HCHs?? Do you mean, not detectable with this analysis; below LOD?

Answer: Yes, we meant below LOD. Corrections in the text have been done.

·         Line 364: present; again, this is dependent on your detection limit.

Answer: Agree. Corrections in the text have been done.

·         Lines 391-392: did you consider possible different kinetics and half-lives in different species?

Answer: Yes, kinetics can play a role, but the information on these processes is insufficient. In our opinion the huge blubber of marine mammals could not be so selective regarding different POPs (all of the studied legacy POPs are lipophilic). Variations in feeding locations could be more influential. 

Reviewer 2 Report

Recommendations

Introduction need major rewrite as the current form is just series of sentences with no flow or coherence message being passed.

Materials and Methods sections need to be rewritten by breaking it into subsections

There is a need of consistency and flow of discussion in the Results and Discussions sections

The authors need to create a separate list of abbreviations

Author Response

Comments and Suggestions for Authors

Recommendations

·         Introduction need major rewrite as the current form is just series of sentences with no flow or coherence message being passed.

Answer: In order to improve the flow we divided the section “Introduction” into three subsections: 1.1. Overview; 1.2. Circumpolar POPs trends in local foods, a brief summary; 1.3. Objectives and tasks of the follow-up study in coastal Chukotka.

·         Materials and Methods sections need to be rewritten by breaking it into subsections.

Answer: The reviewer appears to have overlooked the fact that we have already divided the section “Materials and Methods” into 4 subsections: 2.1. Field sampling;  2.2. POPs analyzed in the collected samples; 2.3. Chemical methods and laboratory equipment used for POPs analysis; 2.4. Processing, analysis and interpretation of the data.

·         There is a need of consistency and flow of discussion in the Results and Discussions sections.

Answer: We did our best to produce a coherent scientific narrative that is consistent throughout (incl. the Results and Discussions sections).

·         The authors need to create a separate list of abbreviations.

Answer: There are relatively few abbreviations (mostly the names of legacy POPs) in the text. Our intent was to define all of the abbreviations at the first instance of their use in the text. We have gone through the text and added the previously missing definitions.

Reviewer 3 Report

General comments

Dudarev et al. present a limited set of measurements of POPs in components of the traditional diet in Chukotka.  These results are compared with results from 15 years previous to this study as well as to literature values for other arctic areas.  The data set, although very limited in numbers of samples of food items, is a valuable addition to the record of monitoring of contaminant exposures to indigenous peoples around the Arctic.  The authors combine the concentration measurements with estimates of diet composition to arrive at contaminant uptake rates that may be compared with health standards as well as used to assess the major routes of exposure.  These goals are certainly appropriate for the audience of this journal.

Although there is value in performing exploratory research, such research is not suitable for publication unless it is of sufficient rigor to substantiate conclusions that are drawn.  One or two samples of a food item or “washout” is not adequate sampling to document the significance of these potential exposure routes.  I recommend that the authors delete from the manuscript the results and discussion of sample types (bird, home-brewed alcohol, insecticide gel, wash-outs) for which there are fewer than three analyses. 

An important component of scientific rigor is quality assurance.  The authors appear to have conscientiously applied good QA protocols, but the results are not well documented.  Given the variable numbers of chlorines on the compounds, it is hard to believe that all would have identical detection limits.  It would be useful to report the range of recovery of the surrogate standards.  Some indication of the precision of analysis would also be good.  It would be nice to know which specific inter-calibration standards were used for QA/QC.  There is no mention of blanks being analyzed; were they?  It would also be helpful to spell out chemicals whose acronyms are not well known (e.g., DBOF, 1234-TKN).

A key piece of information used to calculate consumption rates are the diet components, but these are not reported in a fashion that allows the reader to assess their validity.  We are not told when the survey was performed, the number of surveys sent out, the return rate, or the variability among responses.  If this survey was performed as part of the current study, it should be documented more thoroughly.  If this survey was done as part of another study, that citation should be given, and the appropriateness of the survey to this study (distance in time, adequacy of sampling) should be discussed.

It would be helpful if the authors could more clearly indicate the statistical bases for information that is presented.  Table 3 provides a qualitative comparison of the current study with past data, but we are told (line 288) that some of the “trends” indicated in the table are not statistically significant.  Because the earlier data are not presented, a reader cannot tell which trends are significant and which not.  At a minimum, a star or bold coloration should be used to indicate which trends are significant and which not. 

While I commend the authors on a well organized paper, the language will require considerable improvement prior to publication.

Specific Comments:

1.        Throughout.  It is desirable that consistent units be used throughout the paper.  The authors alternate between using µg/kg w wt, and ng/g which is sometimes designated as l.wt.

2.       Line 31.  It is not accurate to say that POPs are polyhalogenated hydrocarbons; some POPs are not.  It would be accurate to say that many polyhalogenated hydrocarbons are POPs.

3.       Lines 123-131.  References for the material presented in this paragraph should be provided.

4.       Table 1 and text.  Delete categories of measurements for which only a single or pair of measurements exist.  Larger sample numbers are needed to assure reliability of the data being published.

5.       Line 199 and subsequently.  The authors indicate that they measured 15 individual PCB congeners and Sum(PCBs).  If the ΣPCBs is based only on these 15 congeners, it is desirable to indicate that in the notation (e.g., Σ15PCBs or ΣPCB15 as in Table 2).  The true total PCB concentration may be much larger than the sum of these 15 congeners.

6.       Line 209.  Clarify what compound is DBOF.  Indicate the range of recovery of surrogate standards.

7.       Line 225.  Clarify what standard is denoted by 1234-TCN.

8.       Lines 238-245.  More information is needed on the food consumption rates used to calculate contaminant intakes.  Details on the questionnaire should be provided.  It would be useful to tabulate the food consumption rates for comparison with those of other indigenous groups.

9.       Figures 1 and 2.  Put Y-axis labels (Contaminant Concentration, ng/g w w) on the left-most graphs.

10.   Table 2.  Use consistent symbol for decimals (comma vs. period).

11.   Line 301.  What is the basis for the 150 g/meal assumption?  Is this the assumption behind the Russian Allowable Limits?  How does this compare with the indigenous practices?

12.   Lines 535-560, 583-586.  I do not think that the sparse sampling is sufficient to warrant this discussion or the conclusions.

Author Response

Comments and Suggestions for Authors

General comments

Dudarev et al. present a limited set of measurements of POPs in components of the traditional diet in Chukotka.  These results are compared with results from 15 years previous to this study as well as to literature values for other arctic areas.  The data set, although very limited in numbers of samples of food items, is a valuable addition to the record of monitoring of contaminant exposures to indigenous peoples around the Arctic.  The authors combine the concentration measurements with estimates of diet composition to arrive at contaminant uptake rates that may be compared with health standards as well as used to assess the major routes of exposure.  These goals are certainly appropriate for the audience of this journal.

1.     Although there is value in performing exploratory research, such research is not suitable for publication unless it is of sufficient rigor to substantiate conclusions that are drawn.  One or two samples of a food item or “washout” is not adequate sampling to document the significance of these potential exposure routes.  I recommend that the authors delete from the manuscript the results and discussion of sample types (bird, home-brewed alcohol, insecticide gel, wash-outs) for which there are fewer than three analyses. 

Answer: Our study settlements are remote communities that are difficult to reach due to extremely limited transportation options that are also vulnerable to weather. We worked within the logistical and budgetary constraints to collect samples of the key species (as inclusive and diverse as possible) of local fauna and flora to assess the present day contamination of the maximum wide spectrum of species, which are consumed by local people, being the main sources of POPs exposure, and to compare these detected values with previous studies in the region and in neighboring foreign Arctic territories. Under such circumstances each sample (collected and analyzed) is very important for the entire goal and different tasks of the project.

As for the additional domestic sources of food contamination by POPs, each sample of home-brewed alcohol is especially valuable and informative, since nearly all of the home-brewed alcohol consumed in a village is manufactured by just a handful of “producers” in the village. The same goes for insecticide gel, which is delivered to a village in the batches from few manufacturers, and local residents use the same type of insecticide in all houses for a long time (therefore the samples of wash-outs are also important).

2.     An important component of scientific rigor is quality assurance.  The authors appear to have conscientiously applied good QA protocols, but the results are not well documented.  Given the variable numbers of chlorines on the compounds, it is hard to believe that all would have identical detection limits. 

Answer: The level 0.1 ng/g was conditionally adopted as detection limit; at this level all the compounds under study were confidently determined. The real detection limit could vary depending on the sample properties, recovery, etc., but was always not lower than 0.1ng/g.

3.     It would be useful to report the range of recovery of the surrogate standards.  Some indication of the precision of analysis would also be good. 

Answer: Range of recovery for surrogate standards is between 50-110%. If recovery went beyond the specified limits, the sample was analyzed again, that is, a new weight of the specimen was taken and processed (new extraction, concentration, etc), and then the chromatographic analysis was performed again.

4.     It would be nice to know which specific inter-calibration standards were used for QA/QC. 

Answer: inter-calibration was carried out using Canadian Northern Contaminants Program standards (VI, VII and VIII rounds).

5.     There is no mention of blanks being analyzed; were they? 

Answer: Analysis of the “blanks” was performed for each type of species and for each batch of solvents used. The success criterion for blanks was the absence of contamination on the chromatogram, i.e. peaks with the same retention times close to the retention times of the target components.

6.     It would also be helpful to spell out chemicals whose acronyms are not well known (e.g., DBOF, 1234-TKN).

Answer: 4,4’-dibromo-octafluorobyphenyl (DBOF); 1,2,3,4- tetrachloronaphthalene (1,2,3,4-TCN). Both were added in the text.

7.     A key piece of information used to calculate consumption rates are the diet components, but these are not reported in a fashion that allows the reader to assess their validity.  We are not told when the survey was performed, the number of surveys sent out, the return rate, or the variability among responses.  If this survey was performed as part of the current study, it should be documented more thoroughly.  If this survey was done as part of another study, that citation should be given, and the appropriateness of the survey to this study (distance in time, adequacy of sampling) should be discussed.

Answer: the dietary survey was performed as part of the current study, and it is the term of the first article in the series of 4 articles entitled “Traditional Diet and Environmental Contaminants in Coastal Chukotka” submitted to the special issue of the IJERPH. The name of this first article is “Study Design and Dietary Patterns”; it was already officially accepted for the publication. 

8.     It would be helpful if the authors could more clearly indicate the statistical bases for information that is presented.  Table 3 provides a qualitative comparison of the current study with past data, but we are told (line 288) that some of the “trends” indicated in the table are not statistically significant.  Because the earlier data are not presented, a reader cannot tell which trends are significant and which not.  At a minimum, a star or bold coloration should be used to indicate which trends are significant and which not. 

Answer: it was clearly stated (Lines 279-281) that “Due to the limited number of the collected and analyzed samples the present Chukotka study cannot provide a reliable or statistically significant comparison with bigger number of samples collected 15 years ago”. All trends presented in Table 3 are insignificant statistically; only tendencies are depicted.

9.     While I commend the authors on a well organized paper, the language will require considerable improvement prior to publication.

Answer: we will do our best.

Specific Comments:

10.  Throughout.  It is desirable that consistent units be used throughout the paper.  The authors alternate between using µg/kg w wt, and ng/g which is sometimes designated as l.wt.

Answer:  µg/kg ww now is throughout the text; corrections have been done everywhere.

11.  Line 31.  It is not accurate to say that POPs are polyhalogenated hydrocarbons; some POPs are not.  It would be accurate to say that many polyhalogenated hydrocarbons are POPs.

Answer: Agree; correction has been done in the text.

12.  Lines 123-131.  References for the material presented in this paragraph should be provided.

Answer: reference was provided; also quotes were made now.

13.  Table 1 and text.  Delete categories of measurements for which only a single or pair of measurements exist.  Larger sample numbers are needed to assure reliability of the data being published.

Answer: see the answer to item 1.

14.  Line 199 and subsequently.  The authors indicate that they measured 15 individual PCB congeners and Sum(PCBs).  If the ΣPCBs is based only on these 15 congeners, it is desirable to indicate that in the notation (e.g., Σ15PCBs or ΣPCB15 as in Table 2).  The true total PCB concentration may be much larger than the sum of these 15 congeners.

Answer: Agree. ΣPCBs was corrected to ΣPCB15 everywhere where it was needed.

15.  Line 209.  Clarify what compound is DBOF.  Indicate the range of recovery of surrogate standards.

Answer: 4,4’-dibromo-octafluorobyphenyl (DBOF); was added in the text.

16.  Line 225.  Clarify what standard is denoted by 1234-TCN.

Answer: 1,2,3,4- tetrachloronaphthalene (1,2,3,4-TCN); was added in the text.

17.  Lines 238-245.  More information is needed on the food consumption rates used to calculate contaminant intakes.  Details on the questionnaire should be provided.  It would be useful to tabulate the food consumption rates for comparison with those of other indigenous groups.

Answer:  the dietary survey was performed as part of the current study, and it is discussed the first article in the series of 4 articles entitled “Traditional Diet and Environmental Contaminants in Coastal Chukotka” submitted to the special issue of the IJERPH. The name of this first article is “Study Design and Dietary Patterns”; it was already officially accepted for the publication. 

18.  Figures 1 and 2.  Put Y-axis labels (Contaminant Concentration, ng/g w w) on the left-most graphs.

Answer:  µg/kg ww have been inserted on the left-most graphs of Figures 1 and 2.

19.  Table 2.  Use consistent symbol for decimals (comma vs. period).

Answer: corrections have been done; commas put in the Table 2.

20.  Line 301.  What is the basis for the 150 g/meal assumption?  Is this the assumption behind the Russian Allowable Limits?  How does this compare with the indigenous practices?

Answer:  1) It was impossible within our research setting to administer a personalized 24-hour recall food survey. We therefore conducted a broad-based survey on dietary consumption practices, using standard questionnaire. Standard questionnaires are known to be less effective in reflecting the actual consumption practice (particularly in the Arctic), but that was the best available option. 2) We are not nutritionists; we are environmental health researchers. Our main task is to assess the dietary food exposure (to contaminants) of a community, based on the values of local food contamination. 3) The principle to which we adhere over the many years of research in the Russian Arctic is as follows: a thorough assessment of the frequencies of each food item intake is much more informative and reliable as a criterion of real food intakes than an estimation of a portion size; 4) individual fluctuations of the self-reported portion size vary substantially and we need to be able to average these numbers per day/week/month/year; 5) Given that the assumed range of 100g to 200g is used for meat, fish, side dish or salad portions around the world, it is easier and more reliable to use conventionally assumed one portion size as 150g/meal of each foodstuff, and then to average the reported frequencies: 1-3 meals/day; 4-6 meals/week; 1-3 meals/week; 1-3 meals/month; 4-10 meals/year; 1-3 meals/year. This is the reasoning we used in the present study.

21.  Lines 535-560, 583-586.  I do not think that the sparse sampling is sufficient to warrant this discussion or the conclusions.

Answer: see the answer to item 1.